# Non-Abelian three-loop braiding statistics for 3D fermionic topological phases

Jing-Ren Zhou [1,3], Qing-Rui Wang [1,3], Chenjie Wang [2✉] & Zheng-Cheng Gu [1✉]

Fractional statistics is one of the most intriguing features of topological phases in 2D. In particular, the so-called non-Abelian statistics plays a crucial role towards realizing topological quantum computation. Recently, the study of topological phases has been extended to 3D and it has been proposed that loop-like extensive objects can also carry fractional statistics. In this work, we systematically study the so-called three-loop braiding statistics for 3D interacting fermion systems. Most surprisingly, we discover new types of non-Abelian three-loop braiding statistics that can only be realized in fermionic systems (or equivalently bosonic systems with emergent fermionic particles). On the other hand, due to the correspondence between gauge theories with fermionic particles and classifying fermionic symmetry-protected topological (FSPT) phases with unitary symmetries, our study also gives rise to an alternative way to classify FSPT phases. We further compare the classification results for FSPT phases with arbitrary Abelian unitary total symmetry $G^f$ and find systematical agreement with previous studies.

[1] Department of Physics, The Chinese University of Hong Kong, Shatin, New Territories, Hong Kong. [2] Department of Physics and HKU-UCAS Joint Institute for Theoretical and Computational Physics, The University of Hong Kong, Hong Kong, China. [3] These authors contributed equally: Jing-Ren Zhou, Qing-Rui Wang. ✉email: cjwang@hku.hk; zcgu@phy.cuhk.edu.hk

opological phases of quantum matter are a new kind of quantum phases beyond Landau's paradigm. Since the discovery of fractional quantum Hall effect (FQHE), fractionalized statistics of point-like excitations in topological phases has been intensively studied in 2D strongly correlated electron systems. In the past decade, the theoretical prediction and experimental discovery of topological insulator and topological superconductor in 3D systems have further extended our knowledge of topological phases into higher dimensions. As a unique feature, the excitations of 3D topological phases not only contain point-like excitations, but also contain loop-like excitations. Therefore, the fundamental braiding process is not only limited to particle-particle braiding, but is also extended to particle–loop braiding and loop–loop braiding. It is well known that due to topological reasons, point-like excitations in 3D can only be bosons or fermions. In addition, particle-loop braiding can be understood in terms of Aharonov-Bohm effect and loop-loop braiding is equivalent to particle-loop braiding(one can always shrink one of the loops into a point-like excitation). As a result, for long time people thought there was no interesting fractional statistics in 3D beyond the Aharonov–Bohm effect. Surprisingly, a recent breakthrough pointed out that loop-like excitations can indeed carry interesting fractional statistics via the so-called three-loop braiding process[1–7]: braiding a loop $\alpha$ around another loop $\beta$, while both are linked to a third loop $\gamma$, as shown in Fig. 1. Apparently, such kind of braiding process can not be reduced to the particle-loop braiding due to the linking with a third loop. So far, it has been believed that the three-loop braiding process is the most elementary loop braiding process in 3D.

Another natural question would be: whether we can use three-loop braiding process to characterize and classify all possible topological phases for interacting fermion systems in 3D? Recent studies on the classification of topological phases for interacting bosonic and fermionic systems in 3D suggest a positive answer to the above question[8,9]. Basically, it has been conjectured that all topological phases in 3D can be realized by "gauging" certain underlying symmetry-protected topological (SPT) phases[10,11]. For bosonic systems, the "gauged" SPT states are known as Dijkgraaf–Witten gauge theory, and it has been shown (at least for Abelian gauge groups) that three-loop braiding process of their corresponding flux lines can uniquely characterize and exhaust all Dijkgraaf–Witten gauge theories[4]. For fermionic systems, some particular examples with Abelian three-loop braiding process are also studied recently[12]. However, it is still unclear how to understand general cases. On the other hand, it is well known that in low dimensions (up to 3D), the group cohomology theory[13–15] gives rise to a complete classification of bosonic symmetry-protected topological (BSPT) phases for arbitrary finite

unitary symmetry groups. The classification can be generalized to fermionic symmetry-protected topological (FSPT) phases by more advanced constructions[16–25].

In this work, we attempt to systematically understand the three-loop braiding statistics for gauged interacting FSPT systems with general Abelian unitary symmetries. In particular, we discover new types of non-Abelian three-loop braiding statistics that can be only realized in the presence of fermionic particles (accordingly beyond Dijkgraaf–Witten theories). The simplest symmetry group supporting such kind of non-Abelian three-loop braiding process is $\mathbb{Z}_2 \times \mathbb{Z}_8$ or $\mathbb{Z}_4 \times \mathbb{Z}_4$. (More precisely, the corresponding total symmetry groups are $G^f = \mathbb{Z}_2^f \times \mathbb{Z}_2 \times \mathbb{Z}_8$ or $\mathbb{Z}_2^f \times \mathbb{Z}_4 \times \mathbb{Z}_4$ if we also include the fermion parity symmetry $\mathbb{Z}_2^f$.) A simple physical picture describing the corresponding non-Abelian statistics can be viewed as attaching an open Kitaev's Majorana chain onto a pair of linked flux lines ($\mathbb{Z}_2$ and $\mathbb{Z}_8$ unit flux lines for the former case and two different $\mathbb{Z}_4$ unit flux lines for the latter case).

In 1D, it has been shown that a Majorana chain will carry two protected Majorana zero modes on its open ends[26]. In 2D, it is also well known the vortex(anti-vortex) of a p+ip topological superconductor can carry a single topological Majorana zero mode. Thus, it is very natural to ask if flux lines in 3D can also carry topological Majorana zero mode or not. Surprisingly, we find that flux lines carrying topological Majorana zero modes must be linked to each other, as shown in Fig. 2. In contrast, if the loops are unlinked, they can never carry Majorana zero modes. This is simply because one can always smoothly shrink the flux loop into a point like excitation with a single Majorana zero mode on it, which is not allowed in 3D.

The non-Abelian nature of the new type three-loop braiding process we discovered can be understood as the two-fold degeneracy carried by a pair of linked flux lines, and the braiding statistics between two loops that linked with a third loop should be characterized by a unitary 2 by 2 matrix instead of a simple $U(1)$ phase factor. An alternative way to understand the non-Abelian nature of the three-loop braiding statistics is to use the standard dimension reduction scheme to deform the 3D lattice model into a 2D lattice model[27], i.e., by shrinking the $z$-direction to single lattice spacing such that the flux line along the $z$-direction can be regarded as a 2D particle which is exactly the Ising non-Ableian anyon[28] with quantum dimension $\sqrt{2}$. Finally, by explicitly working out all the algebraic constraints of three-loop braiding process for fermionic systems(or equivalently, bosonic systems with emergent fermionic particles), we not only uncover new types of Ising non-Abelian three-loop braiding, but also derive a complete classification of 3D FSPT phases with Abelian unitary total symmetry $G^f$.

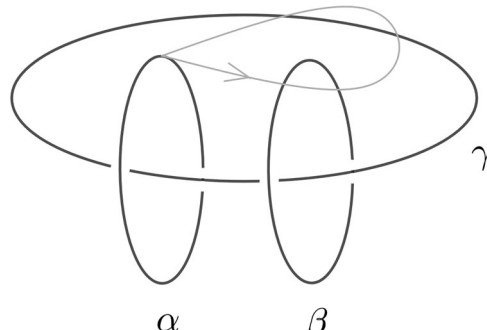

**Fig. 1 Three-loop braiding process.** The three-loop braiding process is braiding one loop $\alpha$ around another loop $\beta$, while both of them are linked to a third loop $\gamma$.

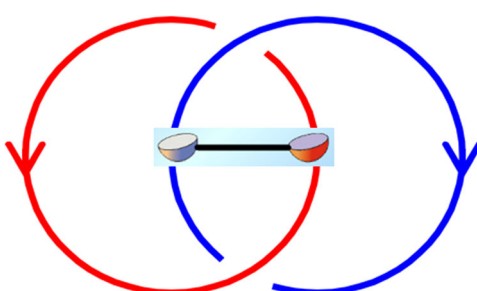

**Fig. 2 The Hopf link of two Ising loops.** Attaching an open Majorana chain onto a pair of linked loops realizes the so-called Ising non-Abelian three-loop braiding process.

## Results

**Symmetries in interacting fermion systems.** We begin with the basics of symmetries in fermionic systems. Fermionic systems have a fundamental symmetry—the conservation of fermion parity: $P_f = (-1)^{N_f}$, where $N_f$ is the total number of fermions. The corresponding symmetry group is denoted as $\mathbb{Z}_2^f = \{1, P_f\}$. In the presence of other global on-site symmetries, the total symmetry group $G_f$ is the central extension of the bosonic symmetry group $G_b$ by the fermion parity $\mathbb{Z}_2^f$, determined by the 2-cocycle $\omega_2 \in H^2(G_b, \mathbb{Z}_2)$. In this work, we consider a general Abelian unitary symmetry group of the following form:

$$G_f = \mathbb{Z}_{N_0}^f \times \prod_{i=1}^{K} \mathbb{Z}_{N_i} \tag{1}$$

where $N_0 = 2m$ is an even integer. One can show that any finite Abelian symmetry group in fermionic systems can be written in this form, after a proper isomorphic transformation.

The bosonic symmetry group is expressed as

$$G_b = G_f / \mathbb{Z}_2^f = \mathbb{Z}_{N_0/2} \times \prod_{i=1}^{K} \mathbb{Z}_{N_i} \tag{2}$$

For simplicity, we will mainly consider the case that $m$ and all $N_i$ are powers of 2, i.e.,

$$N_\mu = 2^{n_\mu}, \quad m = 2^{n_0 - 1} \tag{3}$$

where $n_0 \geq 1$. When $n_0 = 1$ (i.e., $m = 1$), the central extension of $G_b$ is trivial; when $n_0 \geq 2$, the central extension of $G_b$ is nontrivial. This simplification does not exclude any interesting FSPT phases because odd factors of each $N_\mu$ can be factored out. Moreover, $n_2$ and $n_3$ are always trivial if all $N_i$'s are odd integers. Accordingly, neglecting the odd factors, we only lose some BSPT phases, whose classification and characterization are well studied[14].

**Topological excitations and three-Loop braiding in 3D.** Next, we introduce loop braiding statistics in gauged 3D FSPT phase. To study FSPT phases with symmetry group $G_f$, we will gauge the full symmetry. That is, we introduce a gauge field of gauge group $G_f$ and couple it to the FSPT system through the minimal coupling procedure (see refs. [4,10] for details of the procedure). The resulting gauged system is guaranteed to be gapped through that procedure, which is actually topologically ordered. It contains two types of topological excitations:

(i) Point-like excitations that carry gauge charge. We label them by a vector $q = (q_0, \ldots, q_K)$, where $q_\mu$ is an integer defined modulo $N_\mu$. We will use $q$ to denote both the excitation and its gauge charge. This is legitimate because gauge charge uniquely determines charge excitations. Charge excitations are Abelian anyons. Fusing two charge excitations $q_1$ and $q_2$, we obtain a unique charge excitation $q = q_1 + q_2$.

(ii) Loop-like excitations that carry gauge flux. We call them vortices, vortex loops or simply loops, and label them by $\alpha$, $\beta, \ldots$. The gauge flux carried by loop $\alpha$ is denoted by $\phi_\alpha = (\frac{2\pi}{N_0} a_0, \ldots, \frac{2\pi}{N_K} a_K)$, where $a_\mu$ is an integer defined modulo $N_\mu$. There exist many loops that carry the same gauge flux, which differ from each other by attaching charges. Unlinked loops are Abelian, however, they may become non-Abelian when they are linked with other loops. Hence, fusion of vortex loops depend on whether they are linked or not. Nevertheless, regardless Abelian or non-Abelian, gauge flux always adds up. General vortex excitations are not limited to simple loops. For example,

they may be knots or even more complicated structure. In this work, we only consider simple loops and links of them. So far, properties of loops are enough to characterize gauged FSPT systems.

We need to consider three types of braiding statistics between the loops and charges[1]:

First, charge-charge exchange statistics. A charge is either a boson or fermion, depending on the gauge charge it carries. More explicitly, the exchange statistics of charge $q$ is given by

$$\theta_q = \pi q_0 \tag{4}$$

That is, when $q_0$ is odd, it is a fermion. Otherwise, it is a boson. Mutual statistics between charges are always trivial.

Second, charge-loop braiding statistics, which is the Aharonov-Bohm phase given by

$$\theta_{q,\alpha} = q \cdot \phi_\alpha \tag{5}$$

where "$\cdot$" is the vector dot product. We single out a special class of vortex loops, those carrying the fermion parity gauge flux $\phi = (\pi, 0, \ldots, 0)$. We denote these *fermion-parity loops* as $\xi_f$. The mutual statistics between charges and fermion-parity loops are simply given by ref. [29]:

$$\theta_{q,\xi_f} = q \cdot \phi_{\xi_f} = \pi q_0 \tag{6}$$

We notice that the self-exchange statistics of a charge $q$ is equal to Aharonov-Bohm phase $\theta_{q,\xi_f}$, which is required by the very definition of fermion parity symmetry.

Third, loop-loop braiding statistics. It was shown in ref. [1] that the fundamental braiding process between loops is the so-called three-loop braiding statistics (Fig. 1):

Let $\alpha, \beta, \gamma$ be three loop-like excitations. A three-loop braiding is a process that a loop $\alpha$ braids around loop $\beta$ while both linked to a base loop $\gamma$.

On the other hand, if there is no base loop, the two-loop braiding process can always be reduced to charge-loop braiding statistics[1]:

$$\theta_{\alpha\beta} = q_\alpha \cdot \phi_\beta + q_\beta \cdot \phi_\alpha \tag{7}$$

Here $q_\alpha$ is the absolute charge carried by loop $\alpha$, which can be obtained by shrinking the loop to a point. Since charge-loop braiding statistics is universal for all FSPT phases with the same symmetry group $G_f$, two-loop braiding is not able to distinguish different FSPT phases. In the presence of a base loop $\gamma$, the notion of absolute charge is not well defined as shrinking loop $\alpha$ to a point will inevitably touch the base loop. Accordingly, three-loop braiding statistics can go beyond Aharonov-Bohm phases, as already demonstrated in many previous works[1,4,12].

While the gauge group $G_f$ is Abelian, three-loop braiding process is not limited to be Abelian. As mentioned above, linked loops can be non-Abelian in general, and three-loop process involves linked loops. Let us consider loops $\alpha, \beta$, which are linked to the base loop $\gamma$. The base loop $\gamma$ carries gauge flux $\phi_\gamma = (\frac{2\pi}{N_0}, \ldots, \frac{2\pi}{N_K}) \cdot c$, where $c$ is an integer vector. Generally speaking, the fusion space between $\alpha$ and $\beta$, denoted as $V_{\alpha\beta,c}$, is multi-dimensional (we use this notation because the fusion and braiding process only depend on the gauge flux of the base loop). More explicitly,

$$V_{\alpha\beta,c} = \bigoplus_\delta V_{\alpha\beta,c}^\delta \tag{8}$$

where loop $\delta$ are the possible fusion channels of $\alpha$ and $\beta$. Braiding between $\alpha$ and $\beta$ is a unitary transformation in the fusion space, which in general is not just a phase, but a matrix, leading to non-Abelian three-loop braiding statistics. Similarly to anyons in 2D,

**Table 1 Classification of 3D FSPT phases with finite unitary Abelian symmetry groups.**

| Stacking Group | Cases | Classification |
|---|---|---|
| $A$ | If $m$ is odd | $\mathbb{Z}_1$ |
| | If $m$ is even | $\mathbb{Z}_1$ |
| $B_i$ | If $m$ is odd | $\mathbb{Z}_1$ |
| | If $m$ is even | $\mathbb{Z}_{\gcd\{N_0/2,2N_i\}} \times \mathbb{Z}_{\gcd\{N_0/2,N_i\}/2}$ |
| $C_{ij}$ | If $m$ is odd and $N_i = N_j = 2$ | $\mathbb{Z}_2 \times \mathbb{Z}_2$ |
| | If $m$ is odd and $N_i = 2,\ N_j = 4$ | $\mathbb{Z}_4 \times \mathbb{Z}_2$ |
| | If $m$ is odd and $N_i = 2,\ N_j \geq 8$ | $\mathbb{Z}_8 \times \mathbb{Z}_2$ |
| | If $m$ is odd and $4 \leq N_i \leq N_j$ | $\mathbb{Z}_{\gcd\{2N_i,N_j\}} \times \mathbb{Z}_{\gcd\{2N_j,N_i\}} \times \mathbb{Z}_2$ |
| | If $m$ is even | $\mathbb{Z}_{\gcd\{2N_i,N_j\}} \times \mathbb{Z}_{\gcd\{2N_j,N_i\}} \times \mathbb{Z}_{\gcd\{N_0/2,N_{ij}\}} \times \mathbb{Z}_{N_{0ij}/2}$ |
| $D_{ijk}$ | If $m$ is odd and $N_i = N_j = N_k = 2$ | $\mathbb{Z}_2 \times \mathbb{Z}_2$ |
| | If $m$ is odd, $N_i = N_j = 2$ and $N_k \geq 4$ | $\mathbb{Z}_4 \times \mathbb{Z}_2$ |
| | If $m$ is odd and otherwise | $\mathbb{Z}_{N_{ijk}} \times \mathbb{Z}_{N_{ijk}} \times \mathbb{Z}_2$ |
| | If $m$ is even | $\mathbb{Z}_{N_{ijk}} \times \mathbb{Z}_{N_{ijk}} \times \mathbb{Z}_{N_{0ijk}}$ |
| $E_{ijkl}$ | If $m$ is odd | $\mathbb{Z}_{N_{ijkl}}$ |
| | If $m$ is even | $\mathbb{Z}_{N_{ijkl}}$ |

For simplicity, we only consider symmetry groups $\mathbb{Z}_{N_\mu}$ with $N_\mu$ being power of 2, and we assume $N_i \leq N_j \leq N_k \leq N_l$ without loss of generality. $m = N_0/2$ and "gcd" means the greatest common divisor. $N_{ij}$ denotes for the greatest common divisor of $N_i$ and $N_j$, similarly for $N_{0ij}$, $N_{ijk}$, $N_{0ijk}$ and $N_{ijkl}$.

one can define fusion multiplicities $N_{\alpha\beta,c}^{\delta}$, $F$- and $R$-matrices to describe the loop fusion and braiding properties[4]. We give more detailed descriptions in Supplementary Note 1 and 2.

**Classification of FSPT phases via three-loop braiding statistics.** The main purpose of this work is to obtain a classification of 3D FSPT phases via three-loop braiding statistics, and to study non-Abelian three-loop braiding statistics of gauged FSPT phases. We focus on finite Abelian groups of unitary symmetries, which can generally be written as Eq. (1).

We start by defining a set of 3D topological invariants $\{\Theta_{\mu,\sigma}, \Theta_{\mu\nu,\sigma}, \Theta_{\mu\nu\lambda,\sigma}\}$ through the three-loop braiding processes (see "Methods" section for details). Our definitions are very similar to those for 2D FSPTs given in ref. [29], which actually can be related by dimension reduction[4]. Next, we find 14 constraints on $\{\Theta_{\mu,\sigma}, \Theta_{\mu\nu,\sigma}, \Theta_{\mu\nu\lambda,\sigma}\}$, listed in the "Methods" section. Out of these constraints, 7 follow directly from 2D constraints[29], while the other 7 are intrinsically 3D. All intrinsically 3D constraints can be traced back to either the 3D Abelian case[12] or 3D non-Abelian bosonic case[4]. Unfortunately, we are not able to prove all the constraints; those we can prove are discussed in Supplementary Note 3. Finally, by solving the constraints, we obtain a classification of 3D FSPT phases in Table 1. The classification group $H_{\text{stack}}$ under the stacking operation has the following general form:

$$H_{\text{stack}} = A \times \prod_i B_i \times \prod_{i<j} C_{ij} \times \prod_{i<j<k} D_{ijk} \times \prod_{i<j<k<l} E_{ijkl} \quad (9)$$

where $i, j, k, l$ take values in $1, 2, \ldots, K$, and $A, B_i, C_{ij}, D_{ijk}, E_{ijkl}$ are finite Abelian groups. This classification is one of the main results. While it is obtained from a set of partially conjectured constraints, it agrees with all previously known examples. This justifies the validity of the classification. We note that the classification group $A$ is always trivial. However, $A$ is nontrivial for 2D FSPT phases. A newly involved 3D constraint Eq. (55) (see "Methods" section for more details) trivializes it in 3D. Below we discuss more details for the stacking group structure of the classification results.

According to the stacking group Eq. (9) for classifying 3D FSPT phases with Abelian total symmetry $G^f$, we can divide the corresponding topological invariants into five categories, such that the topological invariants in each category are independent

of those in other categories, i.e. the constraints only relate topological invariants inside each category. The five categories are:

(A) $\Theta_{0,0}$, $\Theta_{00,0}$, $\Theta_{000,0}$

(B) (B1) $\Theta_{0,i}$, $\Theta_{00,i}$, $\Theta_{000,i}$

(B2) $\Theta_{i,0}$, $\Theta_{0i,0}$, $\Theta_{ii,0}$, $\Theta_{00i,0}$, $\Theta_{0ii,0}$, $\Theta_{iii,0}$

(B3) $\Theta_{i,i}$, $\Theta_{0i,i}$, $\Theta_{ii,i}$, $\Theta_{00i,i}$, $\Theta_{0ii,i}$, $\Theta_{iii,i}$

(C) (C1) $\Theta_{ij,0}$, $\Theta_{0ij,0}$, $\Theta_{iij,0}$, $\Theta_{jji,0}$

(C2) $\Theta_{ij,i}$, $\Theta_{0ij,i}$, $\Theta_{iij,i}$, $\Theta_{jji,i}$

(C3) $\Theta_{ij,j}$, $\Theta_{0ij,j}$, $\Theta_{iij,j}$, $\Theta_{jji,j}$

(C4) $\Theta_{i,j}$, $\Theta_{0i,j}$, $\Theta_{ii,j}$, $\Theta_{00i,j}$, $\Theta_{0ii,j}$, $\Theta_{iii,j}$

(C5) $\Theta_{j,i}$, $\Theta_{0j,i}$, $\Theta_{jj,i}$, $\Theta_{00j,i}$, $\Theta_{0jj,i}$, $\Theta_{jjj,i}$

(D) (D1) $\Theta_{ij,k}$, $\Theta_{0ij,k}$, $\Theta_{iij,k}$, $\Theta_{jji,k}$

(D2) $\Theta_{jk,i}$, $\Theta_{0jk,i}$, $\Theta_{jjk,i}$, $\Theta_{kkj,i}$

(D3) $\Theta_{ki,j}$, $\Theta_{0ki,j}$, $\Theta_{kki,j}$, $\Theta_{iik,j}$

(D4) $\Theta_{ijk,0}$, $\Theta_{ijk,i}$, $\Theta_{ijk,j}$, $\Theta_{ijk,k}$

(E) $\Theta_{ijk,l}$, $\Theta_{jkl,i}$, $\Theta_{kli,j}$, $\Theta_{lij,k}$

where $A$ is the classification group protected by the symmetry group $\mathbb{Z}_{N_0}^f$, $B_i$ is protected by $\mathbb{Z}_{N_0}^f$ and $\mathbb{Z}_{N_i}$, $C_{ij}$ is protected by $\mathbb{Z}_{N_0}^f$, $\mathbb{Z}_{N_i}$, $\mathbb{Z}_{N_j}$, $D_{ijk}$ is protected by $\mathbb{Z}_{N_0}^f$, $\mathbb{Z}_{N_i}$, $\mathbb{Z}_{N_j}$, $\mathbb{Z}_{N_k}$, and $E_{ijkl}$ is protected by $\mathbb{Z}_{N_i}$, $\mathbb{Z}_{N_j}$, $\mathbb{Z}_{N_k}$, $\mathbb{Z}_{N_l}$.

Mathematically, it is not hard to see that solutions to the constraints form an Abelian group under addition of topological invariants modulo $2\pi$. After all, the constraints are simply a set of homogeneous linear equations. Physically, addition of topological invariants corresponds to stacking of FSPT states, and thereby $H_{\text{stack}}$ is named the "stacking group". To establish the correspondence, we need to show that the topological invariants are indeed additive under stacking operation. We remark that stacking is done on FSPT states before gauging, while the topological invariants are extracted after gauging. Accordingly, stacking additivity is not immediately obvious and deserves some discussion, which we give in Supplementary Note 5. We also note

that the trivial solution (i.e., all topological invariants are zero) to the constraints corresponds to the trivial FSPT state.

We believe the the topological invariants are complete for characterizing FSPT phases with Abelian symmetry group $G_f$ and the constraints are complete so that all solutions are physical. Both completenesses are justified by a comparison with the general group super-cohomology method in Supplementary Note 6.

**Statistics-type indicators.** Our exploration of the classification scheme also uncovers several new kinds of non-Abelian loop braiding statistics, in particular the new kind that involves Majorana zero modes (Fig. 2), which we have briefly mentioned in the introduction. In fact, the correspondence between the layer construction in refs. [23,24] and the three-loop braiding statistics data can be extracted. More explicitly, we pick out several special topological invariants, named statistics-type indicators, to indicate non-Abelian loop braiding statistics with different origins:

(1) $\Theta_{00i,j} = \pi$ ($m$ is odd) is the indicator of the non-Abelian statistics in the Majorana-chain layer, which is generated by the loops carrying unpaired Majorana modes, and a loop carrying one Majorana mode is characterized by its quantum dimension $\sqrt{2}$.

(2) $\Theta_{fi,j} = \frac{N_{0i}}{\gcd(2,N_i)}\Theta_{0i,j} \neq 0$ ($i \neq j$) is the indicator of the complex fermion layer, where "$f$" stands for the fermion-parity loop $\xi_f$ with gauge flux $\phi_{\xi_f} = (\pi, 0, \ldots)$.

(3) $\Theta_{fij,k} = \Theta_{iij,k} = m\Theta_{0ij,k} \neq 0$ is the indicator of the non-Abelian statistics in the complex fermion layer, which is generated by degeneracies in the complex fermion layer and the relevant loops have integer quantum dimension.

(4) $\Theta_{ijk,l} \neq 0$ or $\{\Theta_{fij,k} = 0, \Theta_{0ij,k} \neq 0\}$ is the indicator of the non-Abelian statistics in the BSPT layer, which is generated by degeneracies in the BSPT layer and the relevant loops have integer quantum dimension.

Below we will prove the first statistics-type indicator $\Theta_{00i,j} = \pi$ ($m$ is odd) uniquely indicates the Majorana-chain layer. To proceed, we need to obtain an explicit expression of the topological invariant $\Theta_{\mu\nu\lambda,\sigma}$ as the following (The definitions we used below are introduced in Supplementary Note 2).

We assume that three loops $\xi_\mu, \xi_\nu, \xi_\lambda$ are all linked to a base loop $\xi_\sigma$. Mathematically, let the total fusion outcome $\eta$ of the three loops $\xi_\mu, \xi_\nu, \xi_\lambda$ be fixed, and the standard basis is to let $\xi_\nu$ firstly fuse with $\xi_\mu$, then their fusion channel again fuse with $\xi_\lambda$. We choose the basis of the first local fusion space $V_{\xi_\nu\xi_\mu,c}$ to be diagonalized under the braiding of $\xi_\mu$ around $\xi_\nu$, and the braiding of $\xi_\mu$ around $\xi_\lambda$ is then generally non-diagonalized under this basis, which is expressed as:

$$\widetilde{B}^{\eta}_{\xi_\nu\xi_\mu\xi_\lambda,e_\sigma} = F^{\eta}_{\xi_\nu\xi_\mu\xi_\lambda,e_\sigma} B_{\xi_\mu\xi_\lambda,e_\sigma}(F^{\eta}_{\xi_\nu\xi_\mu\xi_\lambda,e_\sigma})^{-1} :$$
$$\bigoplus_{\delta}(V^{\delta}_{\xi_\nu\xi_\mu,c} \otimes V^{\eta}_{\delta\xi_\lambda,c}) \to \bigoplus_{\rho}(V^{\rho}_{\xi_\nu\xi_\mu,c} \otimes V^{\eta}_{\rho\xi_\lambda,c}) \quad (10)$$

where $\widetilde{B}_{\xi_\nu\xi_\lambda,e_\sigma}$ only braids $\xi_\mu$ around $\xi_\lambda$, while it depends on $\xi_\nu$, as shown in Fig. 3. $B_{\xi_\mu\xi_\nu,e_\sigma}$ is redefined in the same basis as $\widetilde{B}^{\eta}_{\xi_\nu\xi_\mu\xi_\lambda,e_\sigma}$:

$$B_{\xi_\mu\xi_\nu,e_\sigma} : \bigoplus_{\delta}(V^{\delta}_{\xi_\nu\xi_\mu,c} \otimes V^{\eta}_{\delta\xi_\lambda,c}) \to \bigoplus_{\delta}(V^{\delta}_{\xi_\nu\xi_\mu,c} \otimes V^{\eta}_{\delta\xi_\lambda,c}) \quad (11)$$

which has the same expression as $B_{\xi_\mu\xi_\nu,e_\sigma} : \bigoplus_{\delta} V^{\delta}_{\xi_\nu\xi_\mu,c} \to \bigoplus_{\delta} V^{\delta}_{\xi_\nu\xi_\mu,c}$, as though the fusion space is extended, the basis in the extended fusion space should keep diagonalized under the braiding of $\xi_\mu$ around $\xi_\nu$, as shown in Fig. 4. Then $\Theta_{\mu\nu\lambda,\sigma}$ can be expressed

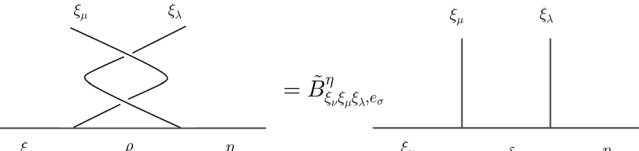

**Fig. 3 The diagrammatic expression of $\widetilde{B}^{\eta}_{\xi_\nu\xi_\mu\xi_\lambda,e_\sigma}$ in the standard basis.** It is braiding of $\xi_\mu$ around $\xi_\lambda$ in the diagonalized basis of braiding $\xi_\mu$ around $\xi_\nu$, which is generally non-diagonalized. $\rho$ is the fusion outcome of $\xi_\mu$ and $\xi_\nu$ in the basis that $\xi_\nu$ and $\xi_\lambda$ are braided. $\delta$ is the fusion outcome of $\xi_\mu$ and $\xi_\nu$ such that $B_{\xi_\mu\xi_\nu,e_\sigma}$ is diagonalized. $\eta$ is the total fusion outcome of $\xi_\mu$, $\xi_\nu$, and $\xi_\lambda$.

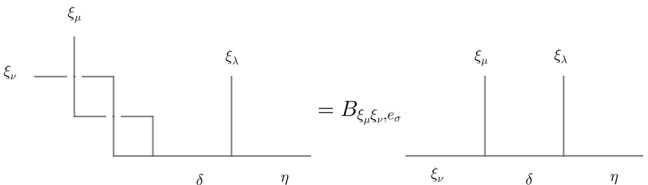

**Fig. 4 The diagrammatic expression of $B_{\xi_\mu\xi_\nu,e_\sigma}$ in the standard basis.** It is braiding of $\xi_\mu$ around $\xi_\nu$ in a basis that we choose to be diagonalized. $\delta$ is the fusion outcome of $\xi_\mu$ and $\xi_\nu$ such that $B_{\xi_\mu\xi_\nu,e_\sigma}$ is diagonalized. $\eta$ is the total fusion outcome of $\xi_\mu$, $\xi_\nu$, and $\xi_\lambda$.

through:

$$e^{i\Theta_{\mu\nu\lambda,\sigma}}I = (\widetilde{B}^{\eta}_{\xi_\nu\xi_\mu\xi_\lambda,e_\sigma})^{-1}(B_{\xi_\mu\xi_\nu,e_\sigma})^{-1}\widetilde{B}^{\eta}_{\xi_\nu\xi_\mu\xi_\lambda,e_\sigma}B_{\xi_\mu\xi_\nu,e_\sigma} \quad (12)$$

where $I$ is the identity matrix in the vector space $\bigoplus_{\delta}(V^{\delta}_{\xi_\nu\xi_\mu,c} \otimes V^{\eta}_{\delta\xi_\lambda,c})$.

Now we are ready to go back to the proof. For simplicity, we can consider $m = 1$ only, which is due to $\mathbb{Z}^f_{2m}$ is isomorphic to $\mathbb{Z}^f_2 \times \mathbb{Z}_m$, and $\mathbb{Z}_m$ can be absorbed into $\prod_i \mathbb{Z}_{N_i}$ part of $G_f$. From constraints Eq. (40) and Eq. (51) in the "Methods" section, $\Theta_{iii,j} = m\Theta_{0ii,j} = m^2\Theta_{00i,j}$. Therefore when $m = 1$, we have the relation $\Theta_{00i,j} = \Theta_{iii,j}$.

Firstly we show that the non-Abelian statistics in Majorana-chain layer (i.e., the Ising type statistics) must have $\Theta_{00i,j} = \pi$: Do a dimension reduction for the gauged Ising type FSPT system from 3D to 2D by choosing $\xi_j$ as the base loop, and condense all the bosonic quasiparticles (as the Ising type statistics is irrelevant to the bosonic matter). The remaining 2D quasiparticles are exactly the Ising anyons: a vortex carrying one majorana mode $\sigma$, a fermion $\psi$ and vacuum 1, which satisfies:

$$e^{i\Theta_{\sigma\sigma\sigma}}I = (\widetilde{B}^{\sigma}_{\sigma\sigma})^{-1}(B_{\sigma\sigma})^{-1}\widetilde{B}^{\sigma}_{\sigma\sigma}B_{\sigma\sigma} = e^{i\pi}\begin{bmatrix} 1 & 0 \\ 0 & 1 \end{bmatrix} \quad (13)$$

where we have $\Theta_{\sigma\sigma\sigma} = \pi$. The 3D topological invariants $\Theta_{iii,j}$ is exactly equal to the 2D one $\Theta_{\sigma\sigma\sigma}$ after dimension reduction and the condensation of all bosons, i.e. $\Theta_{00i,j} = \Theta_{iii,j} = \Theta_{\sigma\sigma\sigma} = \pi$. Secondly, we show that $\Theta_{00i,j} = \pi$ corresponds uniquely to the Ising type statistics: From constraint Eq. (40) in the "Methods" section, when $m = 1$, $\Theta_{00i,j}$ can only take values 0 or $\pi$; when $m$ is even, $\Theta_{00i,j}$ vanishes. We assume the types of non-Abelian statistics in our gauged FSPT system contain only: (1) Ising type in Majorana-chain layer (2) fermionic type in complex fermion layer (3) bosonic type in BSPT layer. Solving the constraints as listed in Supplementary Note 4, and examing the generating phases by mapping to 2D model constructions after dimension reduction[29], we find that to construct the generating phase $\Theta_{00i,j} = \pi$, there always exist loops with quantum dimension $\sqrt{2}$, which is the unique property of Ising anyons in the Majorana-chain layer.

**Table 2 Layer group structure of the classification group of 3D FSPT phases with finite unitary Abelian symmetry groups.**

| | Cases | BSPT phases $\mathcal{B}$ | Cases | Complex fermion $\mathcal{C}$ | Kitaev chain $\mathcal{K}$ | Group structure |
|---|---|---|---|---|---|---|
| $B_i$ | If $m$ is even | $\mathbb{Z}_{\gcd\{N_0/2,N_i\}} \times \mathbb{Z}_{\gcd\{N_0/2,N_i\}/2}$ | If $N_i \geq N_0/2$ | $\mathbb{Z}_1$ | $\mathbb{Z}_1$ | $\mathcal{B}$ |
| | | | If $N_i < N_0/2$ | $\mathbb{Z}_2$ | $\mathbb{Z}_1$ | $\mathcal{C} \ltimes \mathcal{B}$ |
| $C_{ij}$ | If $m$ is odd, $N_i = N_j = 2$ | $\mathbb{Z}_2 \times \mathbb{Z}_2$ | \ | $\mathbb{Z}_1$ | $\mathbb{Z}_1$ | $\mathcal{B}$ |
| | If $m$ is odd, $N_i = 2, N_j = 4$ | $\mathbb{Z}_2 \times \mathbb{Z}_2$ | \ | $\mathbb{Z}_2$ | $\mathbb{Z}_1$ | $\mathcal{C} \ltimes \mathcal{B}$ |
| | If $m$ is odd, $N_i = 2, N_j \geq 8$ | $\mathbb{Z}_2 \times \mathbb{Z}_2$ | \ | $\mathbb{Z}_2$ | $\mathbb{Z}_2$ | $\mathcal{K} \ltimes \mathcal{C} \ltimes \mathcal{B}$ |
| | If $m$ is odd, $4 \leq N_i \leq N_j$ | $\mathbb{Z}_{N_{ij}} \times \mathbb{Z}_{N_{ij}}$ | If $N_i = N_j$ | $\mathbb{Z}_1$ | $\mathbb{Z}_2$ | $\mathcal{B} \times K$ |
| | | | If $N_i < N_j$ | $\mathbb{Z}_2$ | $\mathbb{Z}_2$ | $(\mathcal{C} \ltimes \mathcal{B}) \times K$ |
| | If $m$ is even | $\mathbb{Z}_{N_{ij}} \times \mathbb{Z}_{N_{ij}} \times \mathbb{Z}_{\gcd\{N_0/2,N_j\}} \times \mathbb{Z}_{N_{0ij}/2}$ | If $N_i = N_j$ | $\mathbb{Z}_1$ | $\mathbb{Z}_1$ | $\mathcal{B}$ |
| | | | If $N_i < N_j$ | $\mathbb{Z}_2$ | $\mathbb{Z}_1$ | $\mathcal{C} \ltimes \mathcal{B}$ |
| $D_{ijk}$ | If $m$ is odd, $N_i = N_j = N_k = 2$ | $\mathbb{Z}_2 \times \mathbb{Z}_2$ | \ | $\mathbb{Z}_1$ | $\mathbb{Z}_1$ | $\mathcal{B}$ |
| | If $m$ is odd, $N_i = N_j = 2$ and $N_k \geq 4$ | $\mathbb{Z}_2 \times \mathbb{Z}_2$ | \ | $\mathbb{Z}_2$ | $\mathbb{Z}_1$ | $\mathcal{C} \ltimes \mathcal{B}$ |
| | If $m$ is odd and otherwise | $\mathbb{Z}_{N_{ijk}} \times \mathbb{Z}_{N_{ijk}}$ | \ | $\mathbb{Z}_2$ | $\mathbb{Z}_1$ | $C \times \mathcal{B}$ |
| | If $m$ is even, $N_{0ijk} \neq N_0$ | $\mathbb{Z}_{N_{ijk}} \times \mathbb{Z}_{N_{ijk}} \times \mathbb{Z}_{N_{0ijk}}$ | \ | $\mathbb{Z}_1$ | $\mathbb{Z}_1$ | $\mathcal{B}$ |
| | If $m$ is even, $N_{0ijk} = N_0$ | $\mathbb{Z}_{N_{ijk}} \times \mathbb{Z}_{N_{ijk}} \times \mathbb{Z}_{N_{0ijk}/2}$ | \ | $\mathbb{Z}_2$ | $\mathbb{Z}_1$ | $\mathcal{C} \ltimes \mathcal{B}$ |
| $E_{ijkl}$ | | $\mathbb{Z}_{N_{ijkl}}$ | \ | $\mathbb{Z}_1$ | $\mathbb{Z}_1$ | $\mathcal{B}$ |

We assume $N_i \leq N_j \leq N_k \leq N_l$ without loss of generality. The classification groups of BSPT layer, complex layer and Majorana chain layer are denoted as $\mathcal{B}$, $\mathcal{C}$, and $\mathcal{K}$ respectively. As the group structure depends on more detailed cases beyond Table 1, we list the further cases in column 4. We denote the simple direct product as ×, and non-trivial group extension as ⋉. The classification groups with non-Abelian braiding statistics are denoted in bold italic.

The second statistics-type indicator $\Theta_{fi,j} = \frac{N_{0i}}{\gcd(2,N_i)} \Theta_{0i,j} \neq 0$ $(i \neq j)$ is proposed and proven in ref. [12]. Combining the results in ref. [29] and ref. [12], we infer that $\Theta_{fij,k} = \Theta_{iij,k} = m\Theta_{0ij,k} \neq 0$ is the indicator for the non-Abelian statistics in the complex fermion layer. Finally $\Theta_{ijk,l} \neq 0$ is obviously the indicator for the BSPT layer by the definition of the topological invariant $\Theta_{\mu\nu\lambda,\sigma}$. However, if we consider a special example $G_f = \mathbb{Z}_4^f \times \mathbb{Z}_2 \times \mathbb{Z}_2 \times \mathbb{Z}_2$, where $\Theta_{0ij,k} = \pi$ and $\Theta_{fij,k} = 2\Theta_{0ij,k} = 0$ should still belong to the non-Abelian statistics in BSPT layer. Hence we conclude that $\Theta_{ijk,l} \neq 0$ and $\{\Theta_{fij,k} = 0, \Theta_{0ij,k} \neq 0\}$ are both the indicators for the non-Abelian statistics in BSPT layer.

By checking the linear dependence among the topological invariants, we can also determine relations between the three layers, i.e., simply stacked or absorbed. We summarize the group structure of our classification result by layers, i.e., classification corresponds to BSPT phase, complex fermion layer, and Majorana chain layer, and whether they are non-trivial group extension (we call absorbed) or simple direct product (we call stacking), in Table 2.

Furthermore, invoking the known model construction for 2D FSPT phases[29] and by the fact that quantum dimensions are invariant under dimension reduction, we can find the quantum dimensions of loop-like excitations linked to certain base loops. From the quantum dimensions, we can further show that the non-Abelian three-loop braiding statistics resulting from the Majorana chain layer is due to the unpaired Majorana modes attached to linked loops. Below we will discuss two simplest examples for such kinds of non-Abelian three-loop braiding statistics.

**Example 1 for Ising non-Abelian three-loop braiding statistics:**
$G^f = \mathbb{Z}_2^f \times \mathbb{Z}_2 \times \mathbb{Z}_8$. Firstly, we recall the stacking group classification of FSPT phases:

$$H_{\text{stack}} = A \times \prod_i B_i \times \prod_{i<j} C_{ij} \quad (14)$$

where from Table 1 we know that: $A$ protected by $\mathbb{Z}_2^f$ is trivial, $B_1$ and $B_2$ protected by $\mathbb{Z}_2^f \times \mathbb{Z}_2$ and $\mathbb{Z}_2^f \times \mathbb{Z}_8$, respectively are trivial, while $C_{12}$ protected by $\mathbb{Z}_2^f \times \mathbb{Z}_2 \times \mathbb{Z}_8$ is nontrivial. Therefore the classification of FSPT phases for the symmetry group is $H_{\text{stack}} =$

$C_{12}$. Then we explicitly show the calculation of $C_{12}$: Invoking the known 2D results and combining with the 3D constraints $N_\sigma \Theta_{\mu,\sigma} = 0$, $N_\sigma \Theta_{\mu\nu,\sigma} = 0$, $N_\sigma \Theta_{\mu\nu\lambda,\sigma} = 0$, the generating phases for the subsets (C1), (C2), (C3), (C4) and (C5) are:

$$\left(\Theta_{ij,0}, \Theta_{0ij,0}\right) = \left(\frac{2\pi}{N_{ij}}, 0\right) \times a + \left(0, \frac{2\pi}{N_{0ij}}\right) \times b = (\pi a, \pi b) \quad (15)$$

$$\left(\Theta_{ij,i}, \Theta_{0ij,i}\right) = \left(\frac{2\pi}{N_{ij}}, 0\right) \times c + \left(0, \frac{2\pi}{N_{0ij}}\right) \times d = (\pi c, \pi d) \quad (16)$$

$$\left(\Theta_{ij,j}, \Theta_{0ij,j}\right) = \left(\frac{2\pi}{N_{ij}}, 0\right) \times e + \left(0, \frac{2\pi}{N_{0ij}}\right) \times f = (\pi e, \pi f) \quad (17)$$

$$\left(\Theta_{i,j}, \Theta_{0i,j}, \Theta_{00i,j}\right) = \left(\frac{\pi}{2N_i}, -\frac{\pi}{N_{0i}}, \pi\right) \times g + \left(0, \frac{4\pi}{N_{0i}}, 0\right)$$
$$= \left(\frac{\pi}{4}, -\frac{\pi}{2}, \pi\right) g \quad (18)$$

$$\left(\Theta_{j,i}, \Theta_{0j,i}, \Theta_{00j,i}\right) = \left(\frac{\pi}{N_j}, \frac{2\pi}{N_{0j}}, 0\right) \times N_j h + \left(0, \frac{4\pi}{N_{0j}}, \pi\right) \times i$$
$$= (\pi h, 0, \pi i) \quad (19)$$

where $a, b, c, d, e, f, g, h, i$ are all integers. By the constraint $\Theta_{0ij,0} = \Theta_{00i,j} = -\Theta_{0ij,i} = -\Theta_{00j,i} = -\Theta_{0ij,j}$, we have $b = d = f = g = i$ (mod 2). By the constraint $\Theta_{ij,0} + \Theta_{0j,i} + 4\Theta_{0i,j} = 0$, we have $a = 0$ (mod 2). By the constraint $\Theta_{ij,j} = -4\Theta_{i,j}$, we have $c = -g$ (mod 8). By the constraint $\Theta_{ij,j} = -\Theta_{j,i}$, we have $e = -h$ (mod 2).

Combining all the constraints: $a = 0$ (mod 2), $b = d = f = g = i = -c$ (mod 8), $e = -h$ (mod 2), i.e. the generating phases are:

$$\left(\Theta_{0ij,0}, \Theta_{ij,i}, \Theta_{0ij,j}, \Theta_{0ij,j}, \Theta_{i,j}, \Theta_{0i,j}, \Theta_{00i,j}, \Theta_{00j,i}\right)$$
$$= \left(\pi, \pi, \pi, \pi, \frac{\pi}{4}, -\frac{\pi}{2}, \pi, \pi\right) \quad (20)$$

$$\left(\Theta_{ij,j}, \Theta_{j,i}\right) = (\pi, \pi) \quad (21)$$

while all other topological invariants vanish:

$$\Theta_{0,0} = 0 \quad (22)$$

$$(\Theta_{0,i}, \Theta_{i,0}, \Theta_{0i,0}, \Theta_{00i,0}, \Theta_{i,i}, \Theta_{0i,i}, \Theta_{00i,i})$$
$$= (0, 0, 0, 0, 0, 0, 0) \tag{23}$$

$$(\Theta_{0,j}, \Theta_{j,0}, \Theta_{0j,0}, \Theta_{00j,0}, \Theta_{j,j}, \Theta_{0j,j}, \Theta_{00j,j})$$
$$= (0, 0, 0, 0, 0, 0, 0) \tag{24}$$

$$(\Theta_{ij,0}, \Theta_{0j,i}) = (0, 0) \tag{25}$$

Hence in this case the classification is $\mathbb{Z}_8 \times \mathbb{Z}_2$, which is a $\mathbb{Z}_2$ complex fermion layer absorbed into a $\mathbb{Z}_2 \times \mathbb{Z}_2$ BSPT layer, together forming a $\mathbb{Z}_4 \times \mathbb{Z}_2$ classification, and then a $\mathbb{Z}_2$ Majorana-chain layer again absorbed into the $\mathbb{Z}_4 \times \mathbb{Z}_2$ above, as the complex fermion layer indicator is $\Theta_{fi,j} = \Theta_{0i,j} = -\frac{\pi}{2}$.

Conveniently we can view the "$\mathbb{Z}_8$" part of the classification being generated by:

$$\Theta_{i,j} = \left\{ \frac{\pi}{4}, \frac{\pi}{2}, \frac{3}{4}\pi, \pi, \frac{5}{4}\pi, \frac{3}{2}\pi, \frac{7}{4}\pi, 0 \right\} \tag{26}$$

where $\Theta_{i,j} = \{0, \pi\}$ correspond to Abelian BSPT phases, $\Theta_{i,j} = \{\frac{\pi}{2}, \frac{3}{2}\pi\}$ are Abelian FSPT phases (contain both BSPT layer and complex fermion layer), and $\Theta_{i,j} = \{\frac{\pi}{4}, \frac{3}{4}\pi, \frac{5}{4}\pi, \frac{7}{4}\pi\}$ are non-Abelian FSPT phases (contain all BSPT layer, complex fermion layer and Majorana chain layer). Recall that $\Theta_{00i,j} = \pi$ ($m$ is odd) is the indicator of the Majorana chain layer. The four non-Abelian FSPT phases all have $(\Theta_{00i,j}, \Theta_{00j,i}) = (\pi, \pi)$, which means that loops $\xi_i$ and $\xi_j$ each carry one unpaired Majorana mode simultaneously and both have quantum dimension $\sqrt{2}$, which is the origin of the non-Abelian statistics in Majorana chain layer. On the other hand, the "$\mathbb{Z}_2$" part of the classification is generated by:

$$\Theta_{j,i} = \{0, \pi\} \tag{27}$$

where $\Theta_{j,i} = \pi$ is a non-trivial BSPT phase, and $\Theta_{j,i} = 0$ is a trivial BSPT phase.

We can also understand the 3D braiding statistics by doing a dimension reduction from 3D to 2D and applying the known model construction for 2D generating phases[29]. Firstly we choose $\xi_j$ always to be the base loop, and the 2D system after dimension reduction has symmetry $\mathbb{Z}_2^f \times \mathbb{Z}_2$, which has only one generating phase $(\Theta_i, \Theta_{0i}, \Theta_{00i}) = (\frac{\pi}{4}, -\frac{\pi}{2}, \pi)$, i.e. the subset (C4) in category C. It can be realized by a two-layer model construction: the first layer $a$ is a charge-2 superconductor with chiral central charge $-\frac{1}{2}$ (Ising type), while the second layer $b$ is a charge-2 superconductor with chiral central charge $\frac{1}{2}$ (Ising type). The 2D vortex $\xi_0$ is composited by a unit-flux vortex in layer $a$ and a unit-flux vortex in layer $b$, which therefore has quantum dimension 2. The 2D vortex $\xi_i$ is composited only by a unit-flux vortex in layer $b$, which therefore has quantum dimension $\sqrt{2}$. As the quantum dimensions of loops are invariant under dimension reudction, we conclude that for non-Abelian FSPT phases, with $\xi_j$ all being base loops, loop $\xi_0$ has quantum dimension 2 and loop $\xi_i$ has quantum dimension $\sqrt{2}$.

Secondly, we choose $\xi_i$ always to be the base loop, and the 2D system after dimension reduction has symmetry $\mathbb{Z}_2^f \times \mathbb{Z}_8$, which has two generating phases $(\Theta_j, \Theta_{0j}, \Theta_{00j}) = (\frac{\pi}{8}, \pi, 0)$ and $(\Theta_j, \Theta_{0j}, \Theta_{00j}) = (0, 0, \pi)$, where the first one is trivialized to a $\mathbb{Z}_2$ BSPT in 3D, and both constitute the subset (C5) in category C. Only the second generating phase corresponds to non-Abelian statistics and can be realized by a three-layer model construction: the first layer $a$ is a charge-2 superconductor with chiral central charge $-\frac{1}{2}$ (Ising type), the second layer $b$ is a charge-8 superconductor with chiral central charge 0 (Abelian layer), and the third layer $c$ is a

charge-2 superconductor with chiral central charge $\frac{1}{2}$ (Ising type). The 2D vortex $\xi_0$ is composited by a unit flux in layer $a$, four times of unit flux in layer $b$, and a unit flux in layer $c$, which therefore has quantum dimension 2. The 2D vortex $\xi_j$ is composited only by a unit flux in layer $b$ and a unit flux in layer $c$, which therefore has quantum dimension $\sqrt{2}$.

Thirdly we do not specify the base loop, and let the 2D system after dimension reduction have the full symmetry $\mathbb{Z}_2^f \times \mathbb{Z}_2 \times \mathbb{Z}_8$, which has two generating phases $(\Theta_{ij,0}, \Theta_{0ij,0}) = (\pi, 0)$ and $(\Theta_{ij,0}, \Theta_{0ij,0}) = (0, \pi)$ (or $(\Theta_{ij,i}, \Theta_{0ij,i}) = (\pi, 0)$ and $(\Theta_{ij,i}, \Theta_{0ij,i}) = (0, \pi)$, $(\Theta_{ij,0}, \Theta_{0ij,0}) = (\pi, 0)$ and $(\Theta_{ij,0}, \Theta_{0ij,0}) = (0, \pi)$), i.e. the subset (C1) (or (C2), (C3)) in category C. Only the second generating phase corresponds to non-Abelian statistics and can be realized by a four-layer model construction: the first layer $a$ is a charge-2 superconductor with chiral central charge $-\frac{1}{2}$ (Ising type), the second layer $b$ is a charge-2 superconductor with chiral central charge 0 (Abelian layer), the third layer $c$ is a charge-8 superconductor with chiral central charge 0 (Abelian layer), and the fourth layer $d$ is a charge-2 superconductor with chiral central charge $\frac{1}{2}$ (Ising type). The 2D vortex $\xi_0$ is composited by a unit flux in layer $a$, a unit flux in layer $b$, four times of unit flux in layer $c$, and a unit flux in layer $d$, which therefore has quantum dimension 2. The vortex $\xi_i$ is composited by a unit flux in layer $b$ and a unit flux in layer $d$, which therefore has quantum dimension $\sqrt{2}$. Similarly, the vortex $\xi_j$ is composited by a unit flux in layer $c$ and a unit flux in layer $d$, which also has quantum dimension $\sqrt{2}$. In conclusion, we find that no matter how we do the dimension reduction, the quantum dimensions of the loops coincide, i.e. in our three-loop braiding system with full symmetry $\mathbb{Z}_2^f \times \mathbb{Z}_2 \times \mathbb{Z}_8$, for those non-Abelian FSPT phases, the loop $\xi_0$ has quantum dimension 2, and loops $\xi_i$ and $\xi_j$ both have quantum dimension $\sqrt{2}$, which means that loops $\xi_i$ and $\xi_j$ each carry an unpaired Majorana mode.

**Example 2 for Ising non-Abelian three-loop braiding statistics:** $\mathbf{G}^f = \mathbb{Z}_2^f \times \mathbb{Z}_4 \times \mathbb{Z}_4$. Similarly in the stacking group classification, $A, B_1, B_2$ are all trivial, and we only need to consider $H_{stack} = C_{12}$. Invoking the known 2D results and combining with the 3D constraints $N_\sigma \Theta_{\mu,\sigma} = 0$, $N_\sigma \Theta_{\mu\nu,\sigma} = 0$, $N_\sigma \Theta_{\mu\nu\lambda,\sigma} = 0$, the generating phases for the subsets (C1), (C2), (C3), (C4), and (C5) are:

$$(\Theta_{ij,0}, \Theta_{0ij,0}) = \left(\frac{2\pi}{N_{ij}}, 0\right) \times \frac{N_{ij}}{2} a + \left(0, \frac{2\pi}{N_{0ij}}\right) \times \frac{N_{0ij}}{2} b \tag{28}$$
$$= (\pi a, \pi b)$$

$$(\Theta_{ij,i}, \Theta_{0ij,i}) = \left(\frac{2\pi}{N_{ij}}, 0\right) \times c + \left(0, \frac{2\pi}{N_{0ij}}\right) \times d = \left(\frac{\pi}{2}c, \pi d\right) \tag{29}$$

$$(\Theta_{ij,j}, \Theta_{0ij,j}) = \left(\frac{2\pi}{N_{ij}}, 0\right) \times e + \left(0, \frac{2\pi}{N_{0ij}}\right) \times f = \left(\frac{\pi}{2}e, \pi f\right) \tag{30}$$

$$(\Theta_{i,j}, \Theta_{0i,j}, \Theta_{00i,j}) = \left(\frac{\pi}{N_i}, \frac{2\pi}{N_{0i}}, 0\right) \times 2g + \left(0, \frac{2\pi}{N_{0i}}, \pi\right) \times h \tag{31}$$
$$= \left(\frac{\pi}{2}g, \pi h, \pi h\right)$$

$$(\Theta_{j,i}, \Theta_{0j,i}, \Theta_{00j,i}) = \left(\frac{\pi}{N_j}, \frac{2\pi}{N_{0j}}, 0\right) \times 2l + \left(0, \frac{2\pi}{N_{0j}}, \pi\right) \times m \tag{32}$$
$$= \left(\frac{\pi}{2}l, \pi m, \pi m\right)$$

By the constraint $\Theta_{0ij,0} = \Theta_{00ij,i} = -\Theta_{0ij,i} = -\Theta_{00j,i} = -\Theta_{0ij,j}$, we have $b = d = f = h = m$ (mod 2). By the constraint $\Theta_{ij,0} + \Theta_{0j,i} + \Theta_{0i,j} = 0$, we have $a = 0$ (mod 2). By the constraint $\Theta_{ij,i} = -\Theta_{i,j}$, we have $c = -g$ (mod 4). By the constraint $\Theta_{ij,j} = -\Theta_{j,i}$, we have $e = -l$ (mod 4).

Combining all the constraints: $a = 0$ (mod 2), $b = d = f = h = m$ (mod 2), $c = -g$ (mod 4), $e = -l$ (mod 4), i.e., the generating phases are:

$$(\Theta_{0ij,0}, \Theta_{0ij,i}, \Theta_{0ij,j}, \Theta_{0i,j}, \Theta_{00i,j}, \Theta_{0j,i}, \Theta_{00j,i}) = (\pi, \pi, \pi, \pi, \pi, \pi, \pi) \tag{33}$$

$$(\Theta_{ij,i}, \Theta_{i,j}) = \left(\frac{\pi}{2}, \frac{\pi}{2}\right) \tag{34}$$

$$(\Theta_{ij,j}, \Theta_{j,i}) = \left(\frac{\pi}{2}, \frac{\pi}{2}\right) \tag{35}$$

while all other topological invariants vanish.

Hence in this case the classification is $\mathbb{Z}_4 \times \mathbb{Z}_4 \times \mathbb{Z}_2$, which is a $\mathbb{Z}_4 \times \mathbb{Z}_4$ BSPT simply stacking with a $\mathbb{Z}_2$ "Majorana chain layer absorbed in complex fermion layer", as the complex fermion layer indicator is $\Theta_{fi,j} = \Theta_{0i,j} = \pi$.

The "$\mathbb{Z}_2$" part of the classification can be viewed to be generated by:

$$(\Theta_{0i,j}, \Theta_{00i,j}) = (\pi, \pi) \text{ or } (\Theta_{0j,i}, \Theta_{00j,i}) = (\pi, \pi) \tag{36}$$

while all other $\pi$ valued topological invariants are related by the anti-symmetric constraint of $\Theta_{\mu\nu\lambda,\sigma}$. We do a dimension reduction by always choosing $\xi_j$ as the base loop, and the 2D system has symmetry $\mathbb{Z}_2^f \times \mathbb{Z}_4$. We find that $(\Theta_{0i}, \Theta_{00i}) = (\pi, \pi)$ is exactly the second generating phase for this 2D FSPT system, which can be realized by a three-layer model construction[29]: the first layer $a$ is a charge-2 superconductor with chiral central charge $\frac{3}{2}$ (Ising type), the second layer $b$ is a charge-4 superconductor with chiral central charge $-2$ (Abelian layer), and the third layer $c$ is a charge-2 superconductor with chiral central charge $\frac{1}{2}$ (Ising type). The 2D vortex $\xi_0$ is composed by a unit flux in layer $a$, two times of unit flux in layer $b$, and a unit flux in layer $c$, which, therefore, has quantum dimension 2. The vortex $\xi_i$ is composed by a unit flux in layer $b$ and a unit flux in layer $c$, which therefore has quantum dimension $\sqrt{2}$. As the quantum dimensions of the loops are invariant under dimension reduction, and the symmetry groups of $\xi_i$ and $\xi_j$ are both $\mathbb{Z}_4$ so that it is free to choose which is $\mathbb{Z}_{N_i}$ and which is $\mathbb{Z}_{N_j}$, we conclude that in our gauged 3D FSPT systems, loop $\xi_0$ has quantum dimension 2 and both loop $\xi_i$ and $\xi_j$ have quantum dimension $\sqrt{2}$.

Then we can again check the quantum dimension of loops by doing the dimension reduction without specifying the base loop, and the 2D system has the full symmetry $\mathbb{Z}_2^f \times \mathbb{Z}_4 \times \mathbb{Z}_4$. The second non-Abelian generating phase $(\Theta_{ij,0}, \Theta_{0ij,0}) = (0, \pi)$ (or $(\Theta_{ij,i}, \Theta_{0ij,i})$, $(\Theta_{ij,j}, \Theta_{0ij,j})$) can also be realized by a four-layer construction similarly as in the first example. Then the quantum dimension of $\xi_0$ will still be found as 2, and the quantum dimensions of $\xi_i$ and $\xi_j$ as both $\sqrt{2}$. Therefore in our construction the nontrivial non-Abelian FSPT phase in the $\mathbb{Z}_2$ classification is due to the unpaired Majorana modes attached on $\xi_i$ and $\xi_j$.

## Discussion

In summary, we obtain the classification of 3D FSPT phases with arbitrary finite unitary Abelian total symmetry $G^f$, by gauging the symmetry and studying the topological invariants $\{\Theta_{\mu,\sigma}, \Theta_{\mu\nu,\sigma}, \Theta_{\mu\nu\lambda,\sigma}\}$ defined through the braiding statistics of loop-like excitations in certain three-loop braiding processes and solving the corresponding constraints for these topological invariants. We

further compare this result with the classification obtained by the general group supercohomology theory in ref. [24] and find a systematical agreement. In particular, we can realize any set of allowed values of topological invariants corresponding to a distinguished FSPT phase. Moreover, from several special topological invariants, we can further identify different origins of Non-Abelian three-loop braiding statistics from the corresponding FSPT constructions, i.e., the Majorana chain layer, and complex fermion layer and BSPT layer. Specifically, we argue that the non-Abelian statistics in the Majorana chain layer is due to the unpaired Majorana modes attached on loops.

For future study, it remains unknown how to apply the braiding statistics method to SPT phases with antiunitary symmetry such as the time reversal symmetry, as we do not know how to gauge an antiunitary symmetry. It is expected to generalize the Abelian total symmetry groups $G^f$ to general non-Abelian symmetry groups and have a complete understanding of topological invariants for FSPT phases in 3D. Of course, how to use Non-Abelian three-loop braiding statistics to realize topological quantum computation would be another fascinating future direction. Potential application in fundamental physics was also discussed in ref. [30], it was conjectured that elementary particles could be further divided into topological Majorana modes attached on linked loops and such a scenario naturally explains the origin of three generations of elementary particles.

## Methods

**Definitions of topological invariants**. In this section, we define the topological invariants $\{\Theta_{\mu,\sigma}, \Theta_{\mu\nu,\sigma}, \Theta_{\mu\nu\lambda,\sigma}\}$ through the three-loop braiding statistics. Then, we discuss the 14 constraints on the topological invariants.

Generally speaking, the full set of braiding statistics among particles and loops is very complicated, in particular when the braiding statistics are non-Abelian. Here, we define a subset of the braiding statistics data, which we call topological invariants. They are Abelian phase factors associated with certain composite three-loop braiding processes, and thereby are easier to deal with. Yet, this subset still contains enough information to distinguish all different FSPT phases, as we will show later.

We will define three types of topological invariants, denoted by $\Theta_{\mu,\sigma}$, $\Theta_{\mu\nu,\sigma}$, and $\Theta_{\mu\nu\lambda,\sigma}$, respectively. The definitions are straightforward generalizations of the 2D counterparts given in ref. [29]. To do that, we introduce a notation. Let $\xi_\mu$ be a loop that carries the type-$\mu$ unit flux, i.e., $\phi_{\xi_\mu} = \frac{2\pi}{N_\mu} e_\mu$, where $e_\mu = (0, \ldots, 1, \ldots, 0)$ with the $\mu$-th entry being 1 and all other entries being 0. Then, we define $\Theta_{\mu,\sigma}$, $\Theta_{\mu\nu,\sigma}$ and $\Theta_{\mu\nu\lambda,\sigma}$ as follows. These definitions work for all $N_\mu$, not limited to the special values in Eq. (3).

(i) We define

$$\Theta_{\mu,\sigma} = \bar{N}_\mu \theta_{\xi_\mu, e_\sigma} \tag{37}$$

where

$$\bar{N}_0 = \begin{cases} 2m, & \text{if } m \text{ is even} \\ m, & \text{if } m \text{ is odd} \end{cases}$$

$$\bar{N}_i = \begin{cases} N_i, & \text{if } m \text{ is even} \\ 2N_i, & \text{if } m \text{ is odd} \end{cases}$$

The quantity $\theta_{\xi_\mu, e_\sigma}$ is the topological spin of the loop $\xi_\mu$, when it is linked to another loop $\xi_\sigma$. It is defined as[28]:

$$e^{i\theta_{\xi_\mu, e_\sigma}} = \frac{1}{d_{\xi_\mu, e_\sigma}} \sum_\delta d_{\delta, e_\sigma} \text{tr}(R^\delta_{\xi_\mu, e_\sigma}) \tag{38}$$

where $R^\delta_{\xi_\mu, e_\sigma}$ is the $R$-matrix between two $\xi_\mu$ loops in the $\delta$ fusion channel, and all loops are linked to $\xi_\sigma$ (see Supplementary Note 2 for details).

(ii) We define $\Theta_{\mu\nu,\sigma}$ as the phase associated with braiding $\xi_\mu$ around $\xi_\nu$ for $N^{\mu\nu}$ times, when both are linked to the base loop $\xi_\sigma$. Here, $N^{\mu\nu}$ is the least common multiple of $N_\mu$ and $N_\nu$. In terms of formulas, we have the following expression

$$e^{i\Theta_{\mu\nu,\sigma}} I = (B_{\xi_\mu \xi_\nu, e_\sigma})^{N^{\mu\nu}} \tag{39}$$

where $B_{\xi_\mu \xi_\nu, e_\sigma}$ denotes the unitary operator associated with braiding $\xi_\mu$ around $\xi_\mu$ only once, while both are linked to $\xi_\sigma$, and $I$ is the identity operator. The operator $B_{\xi_\mu \xi_\nu, e_\sigma}$ can be expressed in term of $R$ matrices, and $F$ matrices if needed, once we choose a basis for the fusion spaces.

(iii) We define $\Theta_{\mu\nu\lambda,\sigma}$ as follows. Consider three loops $\xi_\mu, \xi_\nu, \xi_\lambda$ all linked to a base loop $\xi_\sigma$. Then, $\Theta_{\mu\nu\lambda,\sigma}$ is the phase associated with braiding $\xi_\mu$ around $\xi_\nu$ first,

then around $\xi_\lambda$, then around $\xi_\nu$ in opposite direction and finally around $\xi_\lambda$ in opposite direction.

For the topological invariants $\{\Theta_\mu, \Theta_{\mu\nu,\sigma}, \Theta_{\mu\nu\lambda,\sigma}\}$ to be well-defined, we need to show that (1) The corresponding braiding processes indeed lead to Abelian phases and (2) the Abelian phases only depend on the gauge flux of the loops, i.e. independent of charge attachments. The proofs are the same as those for the 2D topological invariants $\{\Theta_\mu, \Theta_{\mu\nu}, \Theta_{\mu\nu\lambda}\}$, so we do not repeat them here and instead refer the readers to ref. [29]. (The only addition for 3D is that one needs to carry the base loop index $\sigma$ in every step of the proofs). The reason that the proofs are identical is that the 3D invariants $\{\Theta_{\mu,\sigma}, \Theta_{\mu\nu,\sigma}, \Theta_{\mu\nu\lambda,\sigma}\}$ can be related to the 2D invariants $\{\Theta_\mu, \Theta_{\mu\nu}, \Theta_{\mu\nu\lambda}\}$ by dimension reduction[4].

**Constraints of topological invariants**. The topological invariants $\{\Theta_{\mu,\sigma}, \Theta_{\mu\nu,\sigma}, \Theta_{\mu\nu\lambda,\sigma}\}$ should satisfy certain constraints. We claim that they satisfy the following 14 constraints, Eqs. (40)–(46) and Eqs. (51)–(57). While we are not able to prove all the constraints, we believe they are rather complete. At least, the solutions to these constraints are all realized in the layer construction of FSPT phases (see Supplementary Note 4). We divide 14 constraints into two groups.

**Group I**: Seven constraints that follow from the 2D counterparts:

$$\Theta_{\mu\mu\nu,\sigma} = \Theta_{\nu\nu\mu,\sigma} = m\Theta_{0\mu\nu,\sigma} \tag{40}$$

$$\Theta_{\mu\nu,\sigma} = \Theta_{\nu\mu,\sigma} \tag{41}$$

$$N_{\mu\nu}\Theta_{\mu\nu,\sigma} = \mathcal{F}(N^{\mu\nu})\Theta_{\mu\mu\nu,\sigma} \tag{42}$$

$$\frac{N_i}{2}\Theta_{ii,\sigma} = \frac{N_{0i}}{2}\Theta_{0i,\sigma} + \left[\frac{N_i}{2}\mathcal{F}(m) + m\mathcal{F}\left(\frac{N_i}{2}\right)\right]\Theta_{00i,\sigma} \tag{43}$$
$$(N_i \text{ is even})$$

$$\Theta_{ii,\sigma} = \begin{cases} 2\Theta_{i,\sigma} + \mathcal{F}(N_i)\Theta_{iii,\sigma}, \text{ if } N_i \text{ is even} \\ \Theta_{i,\sigma}, \text{ if } N_i \text{ is odd} \end{cases} \tag{44}$$

$$\Theta_{00,\sigma} = \begin{cases} 2\Theta_{0,\sigma}, \text{ if } m \text{ is even} \\ 4\Theta_{0,\sigma} + \Theta_{000,\sigma}, \text{ if } m \text{ is odd} \end{cases} \tag{45}$$

$$\begin{cases} \frac{m}{2}\Theta_{0,\sigma} = 0, \text{ if } m \text{ is even} \\ m\Theta_{0,\sigma} + \frac{m^2-1}{8}\Theta_{000,\sigma} = 0, \text{ if } m \text{ is odd} \end{cases} \tag{46}$$

where $\mathcal{F}(N) = \frac{1}{2}N(N-1)$.

The constraints Eq. (40)–Eq. (46) are exactly the 2D fermionic constraints in ref. [29] with a base loop inserted. Since the 3D topological invariants $\{\Theta_{\mu,\sigma}, \Theta_{\mu\nu,\sigma}, \Theta_{\mu\nu\lambda,\sigma}\}$ are related to the 2D ones $\{\Theta_\mu, \Theta_{\mu\nu}, \Theta_{\mu\nu\lambda}\}$ by dimension reduction, the 3D topological invariants satisfy all the 2D constraints.

We briefly explain the meaning of the above constraints. For constraint Eq. (40), firstly we notice a fact that $N$ copies of the topological invariant $\Theta_{\mu\nu\lambda,\sigma}$ are equivalent to do the braiding process for $N$ copies the type-$\mu$ loop, or type-$\nu$, type-$\lambda$ loop, expressed as:

$$N\Theta_{\mu\nu\lambda,\sigma} = \Theta_{[N\xi_\mu]\nu\lambda,\sigma} = \Theta_{\mu[N\xi_\nu]\lambda,\sigma} = \Theta_{\mu\nu[N\xi_\lambda],\sigma} \tag{47}$$

where $[N\xi_\mu]$ means $N$ copies of the type-$\mu$ loop, which can be obtained directly by the definition of $\Theta_{\mu\nu\lambda,\sigma}$. Then by this fact, the expression $m\Theta_{0\mu\nu,\sigma}$ can be rewritten as[29]:

$$e^{im\Theta_{0\mu\nu,\sigma}} = e^{i\Theta_{[m\xi_0]\mu\nu,\sigma}} = e^{i\Theta_{f\mu\nu,\sigma}} \tag{48}$$

where $f$ is the fermion-parity loop. And constraint Eq. (40) illustrates an equivalence $\Theta_{f\mu\nu,\sigma} = \Theta_{\mu\mu\nu,\sigma}$ explicitly proved in the appendix of ref. [29]. Moreover, as the positions of type-$\mu$ and type-$\nu$ loops are symmetric in $\Theta_{f\mu\nu,\sigma}$, the equality can be extended to $\Theta_{\nu\nu\mu,\sigma}$. The constraint Eq. (41) simply points out that the type-$\mu$ and type-$\nu$ loops are symmetric in a three-loop braiding process. The constraints Eq. (42) and Eq. (43) are obtained by rearranging the order of certain braiding processes, where the rearrangements give rise to the non-Abelian phase factors $\Theta_{\mu\mu\nu,\sigma}$ and $\Theta_{00i,\sigma}$. For constraints Eq. (44) and Eq. (45), there are two corollaries relating the type-$\mu$ loop and its anti-loop[29]:

$$\Theta_{\mu\mu,\sigma} + \Theta_{\mu\bar{\mu},\sigma} = \mathcal{F}(N_\mu)\Theta_{\mu\mu\mu,\sigma} \tag{49}$$

$$\Theta_{\mu\bar{\mu},\sigma} = -2N_\mu\theta_{\xi_\mu,e_\sigma} \tag{50}$$

where $\bar{\mu}$ denotes for the anti-loop $\bar{\xi}_\mu$ with gauge flux $\phi_{\bar{\xi}_\mu} = -\phi_{\xi_\mu}$. Combining the two corollaries and inducing the definition of $\Theta_{\mu,\sigma}$ exactly give constraints Eq. (44) and Eq. (45). And the constraint Eq. (46) obtained by demanding the chiral central charge vanishes for FSPT phases.

**Group II**: Seven constraints that are intrinsically 3D:

$$\Theta_{\mu\nu\lambda,\sigma} = sgn(\widehat{p})\Theta_{\widehat{p}(\mu)\widehat{p}(\nu)\widehat{p}(\lambda),\widehat{p}(\sigma)} \tag{51}$$

$$N_{\mu\nu\lambda\sigma}\Theta_{\mu\nu\lambda,\sigma} = 0 \tag{52}$$

$$N_\sigma\Theta_{\mu\nu,\sigma} = 0 \tag{53}$$

$$N_\sigma\Theta_{\mu,\sigma} = 0 \tag{54}$$

$$\Theta_{\mu,\mu} = 0 \tag{55}$$

$$\frac{N^{\mu\nu\sigma}}{N^{\mu\nu}}\Theta_{\mu\nu,\sigma} + \frac{N^{\mu\nu\sigma}}{N^{\nu\sigma}}\Theta_{\nu\sigma,\mu} + \frac{N^{\mu\nu\sigma}}{N^{\sigma\mu}}\Theta_{\sigma\mu,\nu} = 0 \tag{56}$$

$$\frac{N^{\mu\sigma}}{\widetilde{N}_\mu}\Theta_{\mu\sigma,\mu} + \Theta_{\mu\sigma,\mu} = 0 \, (N^{\mu\sigma} \text{ is even}) \tag{57}$$

where $sgn(\widehat{p}) = (-1)^{N(\widehat{p})}$ and $N(\widehat{p})$ is the number of permutations for the four indices $\mu, \nu, \lambda, \sigma$.

The constraints Eq. (51)–Eq. (57) are newly involved 3D constraints (Specially Eq. (52) is a 2D constraint $N_{\mu\nu\lambda}\Theta_{\mu\nu\lambda,\sigma} = 0$ combined with a 3D constraint $N_\sigma\Theta_{\mu\nu\lambda,\sigma} = 0$), which can be traced from 3D bosonic non-Abelian case[4] and 3D fermionic Abelian case[12]. However, we need to prove that these 3D constraints still hold in 3D fermionic non-Abelian case.

Firstly, we argue that the constraints Eqs. (56), (57) proved in Abelian case still hold in non-Abelian case. The constraint Eq. (56) is called the cyclic relation. Imagining that we create $N^{\mu\nu\sigma}$ identical three-loop systems with identical fusion channel and identical total charge. By anyon charge conservation, after braiding and fusion, the total charge should still be $N^{\mu\nu\sigma}Q_{link}$, where $Q_{link}$ is the total charge for a single three-loop system. Then the next step of the proof is similar to the Abelian case[12], where the difference is that the "vertical" fusions may have multiple fusion channels (differ only by charges). But we do not need to care about the charges attached on the resultant loop after fusion, as finally the total charge should still be $N^{\mu\nu\sigma}Q_{link}$, by which we fall into the same result as the proof in ref. [12]. And constraint Eq. (57) is actually the cyclic relation Eq. (56) divided by half on both sides (mod $2\pi$), which then involves fermionic statistics and hence an intrinsic fermionic constraint. It can be argued that it holds in non-Abelian case in a similar manner.

Then we can rigorously prove the constraints Eq. (52)–Eq. (54). The prerequisite to prove them is to assume a 3D "vertical" fusion rule, which naturally gives the linear properties of the topological invariants, explicitly shown in Supplementary Note 3.

However, the constraints Eq. (51) and Eq. (55) are left unproven. For constraint Eq. (51), it is a generalization of the 2D constraint $\Theta_{\mu\nu\lambda} = sgn(\widehat{p})\Theta_{\widehat{p}(\mu)\widehat{p}(\nu)\widehat{p}(\lambda)}$, where the 2D version can be easily proved by a Borromean ring configuration[4]. While here we generalize the totally anti-symmetric property for the indices of $\Theta_{\mu\nu\lambda,\sigma}$ to the base loop. And the constraint Eq. (55) is simply a conjecture, which means that the topological invariant $\Theta_{\mu,\sigma}$ vanishes if the two linked loops fall into the same type.

## Data availability
The complete topological invariants for 3D FSPT phases with Abelian total symmetry $G^f$ can be found in Supplementary Note 4. The corresponding classification results from general group super cohomology theory can be found in Supplementary Note 5.

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

## Acknowledgements
This work is supported by Hong Kong's Research Grants Council (ECS 21301018, GRF No.14306918, ANR/RGC Joint Research Scheme No. A-CUHK402/18).

## Author contributions
Jingren Zhou and Qingrui Wang carried out the calculations; Chenjie Wang and Zhengcheng Gu supervised the project; Jingren Zhou, Chenjie Wang and Zhengcheng Gu wrote the manuscript. Jingren Zhou and Qingrui Wang prepared Supplementary Notes.

## Competing interests
The authors declare no competing interests.
