## [Peer Review File · Nature Communications]

REVIEWER COMMENTS

Reviewer #1 (Remarks to the Author):

In the manuscript "Non-Abelian Three-Loop Braiding Statistics for 3D Fermionic Topological Phases", the authors study the property of 3D fermionic gauge theories obtained by gauging the global symmetry of 3D fermionic SPT phases. In particular, they propose three-loop braiding data intrinsic to fermionic phases. Assuming that the braiding data is sufficiently novel in literature, I think this work will be an important contribution to the theory of fermionic topological ordered phases and should appear in this journal.

Before I recommend to publish this manuscript, I have to clarify which is novel and which is already known about their braiding data. In addition, I have to understand the connection between their braiding invariants and the data for the classification of fermionic SPT phases, since they argue that their braiding invariants diagnose the SPT classification.

1. In the paper by Kapustin and Thorngren arXiv:1701.08264, they discuss the (G-equivariant) 3+1D 2-Ising TQFT, which is the bosonic shadow theory of the 3D fermionic SPT phases with the global G symmetry. Namely, the 2-Ising TQFT is obtained by gauging the fermion parity symmetry of a certain class of 3D fermionic G-SPT phases (beyond Gu-Wen phases with the Kitaev layer $H^2(G, \mathbb{Z}_2)$ incorporated). They point out that there exists a loop-like excitation σ in the 2-Ising TQFT, which has an exotic braiding property described in Fig.3 of their paper. The property is that, when a pair of σ loops constitutes a Hopf link, we have a string of ψ anyons connecting two σ loops. Since the ψ anyons are regarded as the state with odd fermion parity (after fermionization), I am inclined to connect the phenomena with that described in Fig.2 of the manuscript.

Does the data $\Theta_{\{00i,j\}}$ essentially see the property of σ lines of 2-Ising TQFT described in 1701.08264, or does it correspond to the distinct object?

2. I think I have not understood the definition of $\Theta_{\{00i,j\}}$ (generally, $\Theta_{\{\mu \text{ nu } \lambda, \sigma\}}$) correctly. They seem to define $\Theta_{\{\mu \text{ nu } \lambda, \sigma\}}$ via the braiding of μ with ν & λ (below Eq.13), but I cannot see how $\Theta_{\{00i,j\}}$ can have a nontrivial phase when μ is trivial. It would be nice if the authors provide an explanation how $\Theta_{\{00i,j\}}$ can be nontrivial.

3. In page 8 of the manuscript, the authors argue that some braiding data diagnose the SPT classification. For example, the manuscript says that $\Theta_{\{00i,j\}}$ sees the Kitaev-chain layer, $\Theta_{\{fi,j\}}$ sees the complex fermion layer, and so on. Based on this description, I expect that there is a universal formula for computing the classification data $H^2(G, \mathbb{Z}_2)$ (Kitaev layer), $H^3(G, \mathbb{Z}_2)$ (complex fermion layer) at least partially, based on the indicators Θ proposed in the manuscript. However, I could not find such an explicit formula connecting these two objects. I think it would be much better and readable if the author describes the connection between the braiding and SPT classification in a more concrete fashion.

4. Does there exist a braiding data that diagnoses the p -ip layer $H^1(G, \mathbb{Z}_2)$ of the 3D SPT classification?

Reviewer #2 (Remarks to the Author):

This work is a culmination of several lines of works: the general idea of investigating and classifying SPT phases based on gauging the symmetry and studying the braiding statistics of the topological excitations of the resulting topologically ordered systems; the work of Wang, Lin, and Gu in applying this to fermionic SPT phases in 2d; and the work of Wang and Levin studying 3d bosonic SPT phases

through the three-loop braiding statistics, which they then extended to the case of non-abelian statistics. This work puts all of this together to obtain a classification of 3d fermionic SPT phases protected by finite abelian internal symmetry using loop braiding statistics of the gauged system, and obtain the stacking law for these phases. While such classification can also be obtained from the group supercohomology formalism (and indeed the supplementary materials demonstrate this), the present formalism is much more wieldy, at least for the special case of unitary finite abelian symmetry considered here. In particular, the stacking law for the phases is much more readily obtained because the present classification is based on a set of topological invariants which are simply additive under stacking. While some of the constraints on the topological invariants used to derive their results are conjectural, the overall result is convincing as they demonstrate the congruence of their results with known examples and with the group supercohomology formalism.

I would say this work represents an important step in the investigation and classification of 3d topological phases. In addition to giving a classification of fermionic SPT phases protected by finite abelian unitary symmetry, it is also relevant to the study of 3d fermionic topological orders and symmetric-enriched topological phases, as the data describing them are closely related to their un-gauged counterparts, the fermionic SPTs, according to a work by Lan and Wen.

Some suggestions:

- In the stacking group, the abelian group "A" is always trivial (in Table 1), so it does not seem to be doing anything. Is this "A" there because in the 2d case the corresponding group "A" can actually be nontrivial, and you are carrying over the notation/form from the 2d FSPT paper? If so, I think it should be clarified.
- It is not immediately obvious why the topological invariants Θ are additive under stacking. It seems to me that this should be more clearly explained. It also probably helps to state explicitly that we are stacking the phases as fermionic SPTs, instead of the gauged topological orders themselves. It seems like you have an explanation of these things in the 2d paper but not here; the argument in the 2d paper could probably be adapted to the 3d context.
- In Section III.C of the Supplementary Material: equation references are broken.

Reviewer #1 (Remarks to the Author):

Report:

Before I recommend to publish this manuscript, I have to clarify which is novel and which is already known about their braiding data. In addition, I have to understand the connection between their braiding invariants and the data for the classification of fermionic SPT phases, since they argue that their braiding invariants diagnose the SPT classification.

Reply:

We summarize what is known and what is new below:

What is already known:

(1) The concept of braiding statistics and how it classifies SPT phases are already known in previous works (e.g., M. Levin and Z.-C. Gu, Phys. Rev. B 86, 115109 (2012), C. Wang and M. Levin, Phys. Rev. B 91, 165119 (2015)). (2) The concept of the topological invariants and the constraints on the topological invariants, is already known for 2D fermionic systems (C. Wang, C.-H. Lin, and Z.-C. Gu, Phys. Rev. B 95, 195147 (2017)) and 3D fermionic systems with Abelian total symmetry (M. Cheng, N. Tantivasadakarn, and C. Wang, Phys. Rev. X 8, 011054 (2018)).

What is new:

(1) We explore the non-Abelian braiding statistics in 3D interacting fermion systems. In particular, we define the indicator $\Theta_{00i,j}$: $\Theta_{00i,j} = \pi$ indicates the existence of the Hopf link of σ loops, i.e. the existence of a Kitaev chain layer of the corresponding FSPT phases. (2) We list the full classification data of 3D FSPT phases with finite Abelian symmetry groups in Table I, and the group structure by layers, i.e., BSPT layer, complex fermion layer and Kitaev chain layer in table II.

Each set of values of the topological invariants $\{\Theta_{\mu,\sigma}, \Theta_{\mu\nu,\sigma}, \Theta_{\mu\nu\lambda,\sigma}\}$ have a one-to-one correspondence to an FSPT phase. By comparing the calculation by using general group super cohomology theory, we find that the topological invariants proposed in the work indeed give rise to the complete three-loop braiding statistics for a arbitrary Abelian total symmetry group. Therefore, we claim that the three-loop braiding statistics can classify FSPT phases.

Report:

1. In the paper by Kapustin and Thorngren arXiv:1701.08264, they discuss the (G-equivariant) 3+1D 2-Ising TQFT, which is the bosonic shadow theory of the 3D fermionic SPT phases with the global G symmetry. Namely, the 2-Ising TQFT is obtained by gauging the fermion parity symmetry of a certain class of 3D fermionic G-SPT phases (beyond Gu-Wen phases with the Kitaev layer $H^2(G, Z_2)$ incorporated). They point out that there exists a loop-like excitation σ in the 2-Ising TQFT, which has an exotic braiding property described in Fig.3 of their paper. The property is that, when a pair of σ loops constitutes a Hopf link, we have a string of ψ anyons connecting two σ loops. Since the ψ anyons are regarded as the state with odd fermion parity (after fermionization), I am inclined to connect the phenomena with that described in Fig.2 of the manuscript.

Does the data $\Theta_{\{00i,j\}}$ essentially see the property of σ lines of 2-Ising TQFT described in 1701.08264, or does it correspond to the distinct object?

Reply:

The data $\Theta_{\{00i,j\}}$ indeed see the property of Ising σ loop, i.e. the property of non-Abelian braiding, which appears in the Kitaev chain layer. Fig. 2 of this work and Fig. 3 of the Kapustin-Thorngren paper are essentially the same --- a Hopf link of two σ loops. If $\Theta_{\{00i,j\}} = \pi$, it signals that when the unit type- i flux loop is linked to the unit type- j flux loop, both flux loops are σ loops.

An important question is which Gf allows Hopf links of σ loops, i.e., when $\Theta_{\{00i,j\}}$ is allowed to take the value of π . In this work, we give an explicit answer by solving the (partly conjectured) constraints on topological invariants for finite Abelian groups of unitary symmetries. We find that the simplest Gf that allows $\Theta_{\{00i,j\}}$ to be π is $Z_2^f \times Z_2 \times Z_8$ or $Z_2^f \times Z_4 \times Z_4$. The Kapustin-Thorngren paper only gives an implicit answer through a set of highly non-trivial obstruction formulas.

Report:

2. I think I have not understood the definition of $\Theta_{\{00i,j\}}$ (generally, $\Theta_{\{\mu \nu \lambda, \sigma\}}$) correctly. They seem to define $\Theta_{\{\mu \nu \lambda, \sigma\}}$ via the braiding of μ with ν & λ (below Eq.13), but I cannot see how $\Theta_{\{00i,j\}}$ can have a nontrivial phase when μ is trivial. It would be nice if the authors provide an explanation how $\Theta_{\{00i,j\}}$ can be nontrivial.

Reply:

First of all, we would like to clarify that when $\mu=0$, it does not mean that the loop is “trivial”. It means the unit flux associated with the $Z_{\{N\}}^f$ subgroup, or in the simplest case of Z_2^f , it means the fermion parity flux. Second, to see how $\Theta_{\{00i,j\}}$ can be non-trivial, it is enough to consider the 2D case, i.e., ignore the index j . Let us take the example $Gf=Z_2^f \times Z_2$ to illustrate the non-triviality of

Theta_00i. The root state consists of a p+ip superconductor and a p-ip superconductor, and the bosonic Z2 symmetry acts as the fermion parity of the p+ip layer. To be precise, let σ_1 be the fermion parity vortex in the p+ip layer, and σ_2 be the fermion parity vortex in the p-ip layer. Then, a type-0 flux is $\sigma_1\sigma_2$, and a type-1 flux is σ_1 . Now let us consider three anyons: two fermion parity vortices $\sigma_1\sigma_2$ and $\overline{\sigma_1\sigma_2}$ (\sim is used only to distinguish the two vortices) and a Z2 vortex σ_1 . According to the definition, or the 2D version in Wang-Lin-Gu paper (PRB 2017), we have

$$e^{i\Theta_{001}} = B_{\sigma_1\sigma_2,\sigma_1}^{-1} B_{\sigma_1\sigma_2,\overline{\sigma_1\sigma_2}}^{-1} B_{\sigma_1\sigma_2,\sigma_1} B_{\sigma_1\sigma_2,\overline{\sigma_1\sigma_2}},$$

where $B_{a,b}$ means a full counterclockwise braiding of anyon a round anyon b. We see that σ_2 is totally irrelevant. The phase Theta_001 is non-trivial because σ_1 is a non-Abelian anyon. The precise value that $\Theta_{001} = \pi$ can be obtained using the F and R symbols in Kitaev's seminal paper Ann. Phys. 2006.

Report:

3. In page 8 of the manuscript, the authors argue that some braiding data diagnose the SPT classification. For example, the manuscript says that Theta_{00i,j} sees the Kitaev-chain layer, Theta_{fi,j} sees the complex fermion layer, and so on. Based on this description, I expect that there is a universal formula for computing the classification data $H^2(G,Z2)$ (Kitaev layer), $H^3(G,Z2)$ (complex fermion layer) at least partially, based on the indicators Theta proposed in the manuscript. However, I could not find such an explicit formula connecting these two objects. I think it would be much better and readable if the author describes the connection between the braiding and SPT classification in a more concrete fashion.

Reply:

We emphasize that the classification data of $H^2(G,Z2)$ and $H^3(G,Z2)$ are over-complete: given a set of data from $H^2(G,Z2)$ and $H^3(G,Z2)$ as well as $H^4(G,U(1))$, they may not necessarily describe a physical SPT state, because this set of data is subject to the obstruction vanishing conditions (see the Kapustin-Thorngren paper or Wang-Gu paper). If the obstructions do not vanish, the corresponding SPT is not a valid 3D fermionic SPT state. On the other hand, solutions to the constraints of our topological invariants are all physical SPT states. Accordingly, mapping the classification data $H^2(G,Z2)$, $H^3(G,Z2)$, $H^4(G,U(1))$ to our topological invariants, one has to solve the obstruction vanishing conditions first. Appendix B and C in the Supplemental Material have done this job, which as the Referee can see are lengthy. So, we do not think the requested "explicit formula" is easy to obtain. After the obstructed data in $H^2(G,Z2)$ and $H^3(G,Z2)$ are ruled out, one may define quantities for the remaining physical solutions. However, such quantities are just the indicators Theta_{00i,j} and Theta_{fi,j}, as emphasized in our paper. These indicators work only for finite Abelian group, unfortunately. We have added a table on the group structure of our classification and some discussions in the new version. The classification data of complex fermion layer and Kitaev chain layer can be found for

all cases.

Report:

4. Does there exist a braiding data that diagnoses the $p+ip$ layer $H^1(G, \mathbb{Z}_2)$ of the 3D SPT classification?

Reply:

The $p+ip$ layer only appears when there is anti-unitary symmetry, e.g., time-reversal symmetry. But we do not know how to gauge anti-unitary symmetries and define braiding data so far. Therefore, the braiding data of $p+ip$ layer is beyond the scope of this paper.

Reviewer #2 (Remarks to the Author):

Report:

- In the stacking group, the abelian group "A" is always trivial (in Table 1), so it does not seem to be doing anything. Is this "A" there because in the 2d case the corresponding group "A" can actually be nontrivial, and you are carrying over the notation/form from the 2d FSPT paper? If so, I think it should be clarified.

Reply:

We thank referee for pointing out this very subtle detail. We include the classification group "A" because we do not guarantee it is trivial in the beginning. But only when we have the 3D constraint $\Theta_{\mu, \mu} = 0$ and therefore $\Theta_{0, 0} = 0$, we can conclude that "A" is always trivial. We have added a paragraph in the new version to clarify this.

Report:

- It is not immediately obvious why the topological invariants Θ are additive under stacking. It seems to me that this should be more clearly explained. It also probably helps to state explicitly that we are stacking the phases as fermionic SPTs, instead of the gauged topological orders themselves. It seems like you have an explanation of these things in the 2d paper but not here; the argument in the 2d paper could probably be adapted to the 3d context.

Reply:

We thank Referee's very nice suggestion. We have added a paragraph in the new version of the paper.

Report:

- In Section III.C of the Supplementary Material: equation references are broken.

Reply:

We thank referee's very careful reading. We have fixed the problem for the Supplementary Material.

REVIEWER COMMENTS

Reviewer #1 (Remarks to the Author):

I agree that this paper contains sufficiently new materials, in particular about the exploration of braiding properties of sigma loops. They also elucidated the relationship between SPT classification and the such braidings of 3+1d TQFT obtained by gauging symmetries. Their answers to my question were clear and now I support publishing this manuscript wholeheartedly.

Reviewer #2 (Remarks to the Author):

The authors addressed most of my comments satisfactorily, but I still do not see a justification for *why* the topological invariants are additive under stacking. The additive nature is stated but not justified. Can you not give the same reason as in section VI. A of <https://journals.aps.org/prb/abstract/10.1103/PhysRevB.95.195147> ?

Reviewer #1 (Remarks to the Author):

Report:

I agree that this paper contains sufficiently new materials, in particular about the exploration of braiding properties of sigma loops. They also elucidated the relationship between SPT classification and the such braidings of 3+1d TQFT obtained by gauging symmetries.

Their answers to my question were clear and now I support publishing this manuscript wholeheartedly.

Reply:

We thank the referee for reviewing the manuscript again and for recommending publication in Nat. Common.

Reviewer #2 (Remarks to the Author):

Report:

The authors addressed most of my comments satisfactorily, but I still do not see a justification for why the topological invariants are additive under stacking. The additive nature is stated but not justified. Can you not give the same reason as in section VI. A of <https://journals.aps.org/prb/abstract/10.1103/PhysRevB.95.195147?>

Reply:

We thank the reviewer for reviewing the manuscript for a second time. To reply to his/her question, first, we would like to point out that the argument given in Phys. Rev. B 95, 195147(2017) for the additivity of topological invariants under stacking is independent of dimensionality. So, it applies to the current work of 3D systems too. However, the argument there is very intuitive and perhaps is not enough to clarify the reviewer's concerns. After a careful reading of his/her reports, we suspect that the reviewer is perhaps looking for a more detailed and microscopic argument for the additivity of topological invariants, such that he/she can see (1) the additivity is independent of dimensionality and (2) the additivity holds if stacking is done before gauging (rather than after gauging). Below we give such an argument. This argument is adapted into a part of the very last section of the Supplementary Material. The discussion on additivity of topological invariants in the main text is also modified accordingly.

To begin, we briefly revisit the procedure of gauging symmetry, first introduced in PRB 86, 115109 (2012) in the context of SPT phases (one may also consult Appendix A of PRB 91, 165119 (2015) for a general description of the gauging procedure). This procedure applies to both bosonic and fermionic systems. Let H_{SPT} be a lattice Hamiltonian whose ground state is an SPT state associated with onsite unitary symmetries of group G . To gauge the symmetries, H_{SPT} is minimally coupled to a lattice gauge field A of gauge group G :

$$H_{SPT} \Rightarrow \tilde{H}(A) = H_{SPT}(A) + H_{gauge}(A)$$

where $H_{SPT}(A)$ is obtained from H_{SPT} by minimal coupling, and $H_{gauge}(A)$ describes the Hamiltonian of the gauge field itself. The gauging procedure is designed in a way such that $\tilde{H}(A)$ is still energetically gapped and the magnetic flux on every plaquette is a good quantum number (i.e., gauge flux on every plaquette is conserved). Accordingly, eigenstates are always of the form $|\Psi_{SPT}(\phi_A)\rangle \otimes |\phi_A\rangle$, where ϕ_A is a configure of magnetic flux on the lattice, $|\Psi_{SPT}(\phi_A)\rangle$ describes the state of the degrees of freedom of the original SPT system in the presence of ϕ_A , and $|\phi_A\rangle$ describes the state of the gauge field. $|\Psi_{SPT}(\phi_A = 0)\rangle$ describes eigenstates of the un-gauged SPT system. We remark that this form of eigenstates is not essential but simplifies our following discussions. We also note that the gauge field in this system is almost classical, in the sense that it has no quantum fluctuation at all. Accordingly, its effect is equivalent to a non-dynamical background gauge field, which is also commonly used in the study of topological response theory of SPT systems. However, for the purpose of having well-defined braiding statistics for the vortices, we treat the gauge field quantum mechanically and dynamically.

The topological invariant Θ (any of those defined in the manuscript) is defined as the Berry phase associated with certain cyclic adiabatic process in the presence of magnetic flux: $|\Psi_{SPT}(\phi_A)\rangle_t \otimes |\phi_A\rangle_t$, where t adiabatically changes from 0 to T . Under the assumption that the Berry phase is Abelian (which is true for our topological invariants), we have

$$\begin{aligned}
|\Psi_{SPT}(\phi_A)\rangle_T \otimes |\phi_A\rangle_T &= e^{i\Theta_{SPT}} |\Psi_{SPT}(\phi_A)\rangle_0 \otimes e^{i\Theta_{gauge}} |\phi_A\rangle_0 \\
&= e^{i(\Theta_{SPT} + \Theta_{gauge})} |\Psi(\phi_A)\rangle_0 \otimes |\phi_A\rangle_0
\end{aligned} \tag{1}$$

where Θ_{SPT} comes from the SPT state $|\Psi_{SPT}(\phi_A)\rangle$ and Θ_{gauge} comes from the state of gauge field $|\phi_A\rangle$. [In the expression (1), dynamical phases are neglected and only Berry phases are kept.] An important technical point is that $\Theta_{gauge} = 0$ in the gauging procedure of PRB 86, 115109 (2012), and in our work we have implicitly assume it. In general, Θ_{gauge} can be nonzero if the gauge field involves, say a Chern-Simon-type interaction in $H_{gauge}(A)$. However, it is a natural choice to let the gauge field itself have no topological interaction, such that topological properties are solely from the SPT part. After all, the gauge field is used as a probe here. Accordingly, we have topological invariant $\Theta = \Theta_{SPT}$. We remark that this choice leads to the fact the $\Theta = 0$ if the SPT is trivial. If $\Theta_{gauge} \neq 0$, then the topological invariant $\Theta = \Theta_{gauge}$ for the trivial SPT state which is a bit harder to work with.

With the above clarification, we now consider stacking. Consider two SPT systems, described by H_a and H_b . The stacked system is then described by $H_a + H_b$. After gauging, it is described by

$$H_a + H_b \Rightarrow \tilde{H}_{a+b}(A) = H_a(A) + H_b(A) + H_{gauge}(A)$$

The eigenstates can be denoted by $|\Psi_{SPT}^a(\phi_A)\rangle \otimes |\Psi_{SPT}^b(\phi_A)\rangle \otimes |\phi_A\rangle$. Accordingly, for the same adiabatic process as above, the topological invariant in the stacked system is given by

$$\Theta_{a+b} = \Theta_{SPT}^a + \Theta_{SPT}^b + \Theta_{gauge} = \Theta_{SPT}^a + \Theta_{SPT}^b = \Theta_a + \Theta_b$$

where Θ_{SPT}^a is associated with $|\Psi_{SPT}^a(\phi_{A(t)})\rangle$, Θ_{SPT}^b is associated with $|\Psi_{SPT}^b(\phi_{A(t)})\rangle$, and $\Theta_{gauge} = 0$ is associated with $|\phi_{A(t)}\rangle$, and $\Theta_a = \Theta_{SPT}^a$, $\Theta_b = \Theta_{SPT}^b$ are the topological invariants before stacking. This completes our argument for the additivity of the topological invariants under stacking.

In the above argument, it is obvious that dimensionality is irrelevant. Also, stacking is done before gauging. We think what might have troubled the referee is the implicit assumption $\Theta_{gauge} = 0$. It is the key to make the additivity true for stacking before gauging (additivity for stacking after gauging is obvious and is always true regardless of the type of gauge-field Hamiltonian --- this is another point that the referee wanted us to clarify in the previous report). This assumption is taken in all works that use the method of gauging symmetry in the field of SPT phases, including the very first paper Phys. Rev. B 86, 115109 (2012). We hope that our argument is clear to the reviewer and hope that he/she is satisfied with it.